# Perivascular Epithelioid Cell Tumor (PEComa) of the Pancreas in a Patient with Ulcerative Colitis: A Case Report and Review of the Literature

**DOI:** 10.3390/healthcare11040547

**Published:** 2023-02-12

**Authors:** Maryam A. Almousa, Yara A. Alnashwan, Samir S. Amr

**Affiliations:** 1Department of Pathology and Laboratory Medicine, King Fahad Specialist Hospital, Dammam 34258, Saudi Arabia; 2Department of Pathology, College of Medicine, Imam Abdulrahman Bin Faisal University, Dammam 1982, Saudi Arabia; 3Department of Pathology and Laboratory Medicine, Istishari Hospital, Amman 840431, Jordan

**Keywords:** perivascular epithelioid cell tumors, ulcerative colitis, pancreas, tuberous sclerosis complex

## Abstract

Perivascular epithelioid cell tumors (PEComas) are mesenchymal tumors of peculiar cells that are focally associated with blood vessels, and generally have a distinctive bi-phenotypic expression of both smooth muscle and melanocytic markers. There are several entities in the PEComa family, including tumors that arise in the soft tissues and viscera. Frequently affected organs include the lungs (sugar tumors), uterus, broad ligament, colon, small bowel, liver, and pancreas. Ulcerative colitis (UC) has been associated with the development of tumors, especially colorectal and hepatobiliary carcinomas. Rare cases of UC have been reported in the PEComa family of tumors, but none in the pancreas. Here, we present a case study of a 27-year-old female patient with a history of UC who developed PEComa of the pancreas, a unique association that has not been previously reported. We also review reported cases of PEComas in the pancreas, as well as PEComas at all anatomic sites associated with UC.

## 1. Introduction

The concept of perivascular epithelioid cell (PECs) was introduced by Bonetti et al. (1992) as a novel cell type that was identified in renal angiomyolipoma (AML), clear cell “sugar” tumor of the lung (CCSTL), and pulmonary lymphangioleiomyomatosis (PLAM). They stated that PEC is an unusual cell type that is immunoreactive, with melanocytic markers, particularly HMB45, and exhibits an epithelioid appearance, a clear-acidophilic cytoplasm, and a perivascular distribution [1]. Furthermore, Bonetti et al. also documented a comparative study of three cases each of CCSTL and AML of the kidney. Morphological analysis showed that the CCSTL cells were identical to the perivascular epithelioid component of AML and expressed the melanosome marker HMB45. They concluded that both lesions belonged to the same family of lesions [2]. In a subsequent review article, these authors introduced PEC as a unifying concept for a family of tumors, including AML and its variants, PLAM, and pulmonary and extrapulmonary clear cell “sugar” tumors (CCSTs) [3]. The term PEComa was introduced by Zamboni et al. to describe a pancreatic tumor composed purely of the perivascular epithelioid cell component of this family of tumors [4]. Folpe et al. added another novel entity to the family of PEComa, namely clear cell melanocytic tumors of the falciform ligament and ligamentum teres of the liver [5]. As the name of this tumor implied, it showed immunohistochemical positivity for melanocytic marker (HMB45) and smooth muscle marker (smooth muscle actin).

PEComas have thereafter been reported in many anatomic sites, including the uterus [6], gastrointestinal tract [7,8], liver [9], soft tissue [10], bone [11], and skin [12]. Twenty-eight cases of pancreatic PEComas have been reported [4,13,14,15,16,17,18,19,20,21,22,23,24,25,26,27,28,29,30,31,32,33,34,35,36,37,38], mostly benign, without associated pancreatic or gastrointestinal disease. Herein, we report a unique case of pancreatic PEComa associated with ulcerative colitis, along with a review of the literature.

## 2. Case Report

### 2.1. Clinical History

A 27-year-old non-smoker female with a known long-standing (11 year) history of ulcerative colitis, and maintained on mesalazine, 1000 mg PO TID, with regular follow up, was referred to our hospital, a tertiary care center, in April 2013, from a peripheral hospital where she was found to have a pancreatic tail mass on a computed tomography (CT) scan. Upon presentation at our hospital, she stated that she had an episode of acute severe central abdominal pain that started one month earlier, associated with vomiting, which led her to seek medical attention. She denied a history of fever, weight loss, loss of appetite, or recent changes in bowel habits. Physical examination and laboratory investigations were unremarkable.

On a repeat of CT-scans of the abdomen, pelvis, and chest, a well-defined, solid, oval-shaped, exophytic mass projecting at the pancreatic tail inferiorly, measuring 4 cm in maximum dimension, was found (Figure 1). The mass was iso-attenuated with the pancreatic parenchyma. There was no observable fat plan involvement that could suggest infiltration. No other remarkable lesions, apart from a subserosal uterine fibroid and small bilateral fibroadenomas of the breast, were found on chest, abdominal, and pelvic CT scans.

### 2.2. Pathological Findings

On gross examination, the resected pancreas revealed a solid well-circumscribed, encapsulated, homogenous dark red mass, measuring 4.2 × 4.1 × 3 cm, involving the medial margin. The rest of the pancreas was unremarkable. The spleen weighed 130 g and was unremarkable. Microscopically, the tumor was encapsulated, and the cells were arranged in sheets, with vague nodularity, associated with high vascularity (Figure 2a). The tumor cells were mainly spindle-shaped, with a component of epithelioid cells featuring clear to eosinophilic granular cytoplasm, and centrally located rounded uniform nuclei with inconspicuous nucleoli (Figure 2b,c). They were centered around the blood vessels. Necrosis, significant mitosis, vascular invasion, and atypia were absent. The tumor reached the medial margin of the resection.

Immunohistochemical studies showed co-expression for HMB-45 (Figure 3a), smooth muscle actin (SMA) (Figure 3b), and focal positivity for vimentin. CD34 stain showed rich vascularity with many capillaries closely related to tumor cells (Figure 3c). Tumor cells were negative for pan-cytokeratin (Figure 3d), synaptophysin, neuron specific enolase (NSE), CD34, CD117, Dog-1, and INI-1. Based on the distinctive morphological and immunohistochemical features, PEComa of the pancreas was diagnosed.

The patient was followed up at our outpatient clinic for seven years, and there was no evidence of tumor recurrence despite the positive medial margin of resection. However, the patient was admitted several times to our hospital because of exacerbations of chronic inflammatory bowel disease. She was evaluated retrospectively for tuberous sclerosis complex (TSC), but did not show signs of this complex either clinically or radiologically.

## 3. Discussion

In the classification of tumors of the soft tissue and bone, the World Health Organization (WHO) has defined PEComas as mesenchymal tumors composed of distinctive cells that show a focal association with blood vessel walls, and usually express melanocytic and smooth muscle markers. The PEComa family includes AML, CCST, lymphangioleiomyomatosis (LAM), and a group of histologically and immunophenotypically similar tumors arising at a variety of soft tissue and visceral sites. PEComas other than AML and LAM are rare, with approximately 200 cases reported as of 2013. PEComas are more frequent in females, with a male to female ratio of 1:6 [39]. A subset of PEComas show aggressive behavior. In a study of 26 cases of gynecologic and soft tissue PEComas and a review of 61 additional cases from the literature, Folpe et al. proposed a three-tier classification; namely, benign, of uncertain malignant potential, and malignant. They postulated the following “worrisome” features to differentiate between these categories: size > 5 cm, infiltrative margins, high nuclear grade, cellularity, mitotic rate > 1/50 HPF, necrosis, and vascular invasion. The presence of two or more of these features classifies the tumor as malignant [10]. The tumor in our patient did not exhibit any criteria suggestive of malignancy or malignant potential.

Twenty-nine cases of PEComa of the pancreas, including the current case, have been recorded (Table 1). There were 24 females and 5 males, with a male to female ratio of 1: 4.8. The patients had a mean age of 47.6 (range, 17–74) years. Of the 28 tumors, 4 (14.3%) were malignant [19,23,32,35]. Abdominal or epigastric pain was the most frequent clinical presentation, observed in 16 patients (57.1%). Other presentations included diarrhea, weight loss, melena, and anemia. Interestingly, in five patients (17.8%), tumors were incidentally diagnosed during abdominal radiological studies [21,25,29,30,36].

The tumors were located in the head of the pancreas (*n* = 11), body (*n* = 8), tail (*n* = 4), uncinate process (*n* = 2), isthmus (*n* = 1), head and body (*n* = 1), and in an ectopic pancreatic tissue in the liver (*n* = 1). Tumor size ranged from 0.12 to 11.5 cm, with an average of 3.58 cm. The size of malignant tumors averaged 6.75 (range 4–11.5) cm.

Fine needle aspiration cytology and biopsy (FNAC and FNAB) were performed using endoscopic ultrasound (EUS) guidance in 19 patients (65.5%). In four patients, no surgical resection was performed [33,34,35], including one malignancy that was treated using sirolimus, an mTOR inhibitor [35]. The role of EUS-guided FNAC has been emphasized in several studies [14,22,24,31,34,36]. Sangiorgio et al. reported a series of three hepatic and two pancreatic PEComas diagnosed preoperatively using US-guided FNA. All patients were females (aged 28–70 years), had no history of tuberous sclerosis complex (TSC), and presented with a single, localized, painless mass. Rapid on-site evaluation (ROSE) of cytological samples was performed in all cases to evaluate for cellular content and adequacy of specimens. Direct smear and cell block preparations revealed sheets and nests of epithelioid cells with eosinophilic or vacuolated cytoplasm. Immunohistochemical staining showed these neoplastic cells co-expressing both melanocytic and smooth muscle markers. The study indicated that a correct diagnosis could be achieved with the help of immunohistochemistry and good quality cytological samples obtained via FNA. This is particularly important for planning an adequate surgical strategy and avoiding overtreatment [34].

In 24 patients, surgical procedures were performed to excise the tumors. Depending on the location of the tumor, a variety of surgical procedures were performed; for example, distal pancreatectomy (DP) was performed in eight patients, including splenic-preserving distal pancreatectomy (SPDP; *n* = 1), middle pancreatectomy (MP; *n* = 1), pancreaticoduodenectomy (PD; *n* = 6), and pylorus-preserving pancreaticoduodenectomy (PPPD; *n* = 5). In five patients, the exact surgical procedure was not specified (Table 1).

Follow-up data were available for 23 patients for durations ranging from 3 to 144 months, with an average of 27.3 months. Of the four cases of malignant PEComa, two had developed liver metastases at 39 months [19] and 6 months [23] follow-up. The third patient had no metastasis 13 months after receiving chemotherapy [32]. The fourth patient had a marked reduction in the size of the tumor 42 months after treatment with sirolimus [34].

Of the members of the PEComa family of tumors, renal AML is highly associated with tuberous TSC, a group of autosomal dominant genetic disorders caused by germ-line mutations in TSC1 or TSC2, located on chromosomes 9q and 16p, respectively, which encode the proteins hamartin and tuberin, respectively. In addition to AML, patients with TSC frequently develop symptomatic lesions in the central nervous system, including cerebral cortical tubers, as well as distinctive skin lesions, cardiac rhabdomyomas, and LAM, another member of the PEComa family [40]. However, of the 28 cases of pancreatic PEComas, only one had stigmata of TSC. In this unique case, the patient was clinically diagnosed with TSC at 6 months of age when she developed infantile spasms. Brain MRI showed multiple cortical tubers and subependymal nodules. At 8 years of age, she underwent resection of a subependymal giant cell astrocytoma. Facial angiofibroma, hypopigmented macules, and an ungual fibroma were also identified, providing further evidence of TSC. Bilateral renal AML necessitated radiographic surveillance studies. In 2006, a CT scan with contrast identified an 8-mm cyst in the body of the pancreas; it enlarged to 5.8 cm by 2013. Distal pancreatectomy was performed, and pathological examination showed a 6.5 × 4.5 × 4.5 cm oligolocular cyst in the tail of the pancreas that was histologically diagnosed as a mucinous cystic neoplasm (MCN) with low-grade dysplasia. Three well-circumscribed lesions, measuring 0.6, 0.2, and 0.12 cm, were also identified, and confirmed to be PEComas. Additionally, four well-circumscribed lesions, measuring 0.9, 0.25, 0.1, and 0.05 cm, were identified and confirmed to be neuroendocrine tumors (NET). Such a unique association of pancreatic epithelial MCN, multiple PEComas, and multiple NETs has not been previously reported [29].

In a review of the histology and genetics of PEComas, Thway and Fisher emphasized the mesenchymal nature of these tumors, which are composed of distinctive epithelioid or spindle cells that are immunoreactive for both melanocytic and smooth muscle markers. They stated that PEComas have a spectrum of behavior and outcome ranging from benign to malignant, and histological criteria have been proposed to assess malignant potential. Their differential diagnoses include carcinomas, smooth muscle tumors, clear cell neoplasms, and fatty neoplasms. They constitute a genetically diverse group that includes neoplasms harboring TEF3 gene rearrangements and those with TSC2 mutations, indicating alternative tumorigenic pathways. This can influence future management protocols [41].

Our patient had long-standing ulcerative colitis (UC). It is well known that UC can be associated with the development of colorectal dysplasia and adenocarcinoma [42] with prevalence rates of 4% and 7%, respectively [43]. UC is also associated with primary sclerosing cholangitis, which can predispose a patient to dysplasia that can progress to hepatobiliary carcinoma [44]. Additionally, iatrogenic Kaposi sarcoma and lymphoma can occur in patients with UC following treatment with immunosuppressive drugs [45,46].

Over 60 published inflammatory bowel disease (IBD) susceptibility loci have been discovered and replicated, of which approximately a third are associated with both ulcerative colitis and Crohn’s disease (CD), although 21 are specific to UC and 23 to CD. Notably, genes implicated in mucosal barrier function (ECM1, CDH1, HNF4a, and laminin B1) confer risk of UC; furthermore, E-cadherin is the first genetic correlation between colorectal cancer and UC [47]. These genes have not been observed in PEComas.

We searched the literature for cases of UC associated with tumors of the PEComa family. We found four tumors, and our case was the fifth. These cases are presented in Table 2. They included two liver tumors, one kidney tumor, and one tumor in the cecum, in addition to the pancreatic tumor reported herein [34,48,49,50]. All five tumors, except the present case, were labelled as AML. We speculate that the association of UC and PEComas could be a coincidental fortuitous occurrence, without any shared genetic susceptibility or underlying etiology.

## 4. Conclusions

In conclusion, we have reported a rare case of pancreatic PEComa of the pancreas in a young female with a history of ulcerative colitis. The tumor was resected, and the patient was followed up for seven years with no evidence of recurrence.

## Figures and Tables

**Figure 1 healthcare-11-00547-f001:**
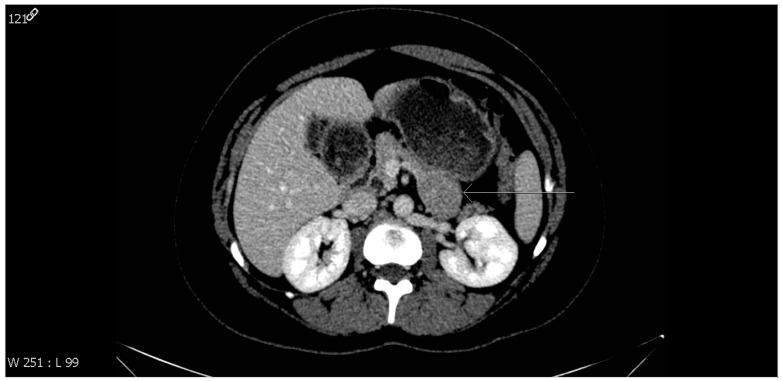
CT scan image of the abdomen showing an oval-shaped homogenous mass arising from the pancreatic tail inferiorly (arrow), measuring approximately 4 cm in diameter. Laparoscopic corporeo-caudal pancreatectomy and splenectomy were performed for complete resection of the mass. No metastatic deposits or nodal enlargements were noted during surgery.

**Figure 2 healthcare-11-00547-f002:**
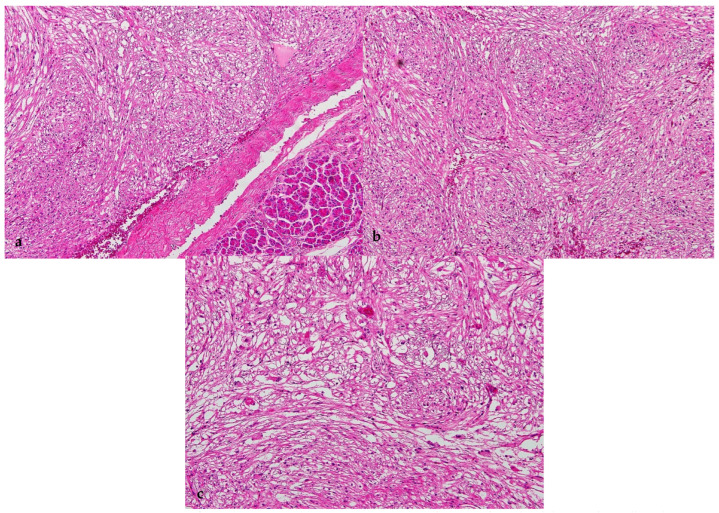
Histological features. Spindle cell tumor surrounded by fibrous capsule separating it from adjacent normal pancreatic acini ((**a**), 100×), the tumor showed nodularity with spindly and epithelioid cells featuring clear to eosinophilic granular cytoplasm and rounded uniform nuclei ((**b**), 100×; (**c**), 200×).

**Figure 3 healthcare-11-00547-f003:**
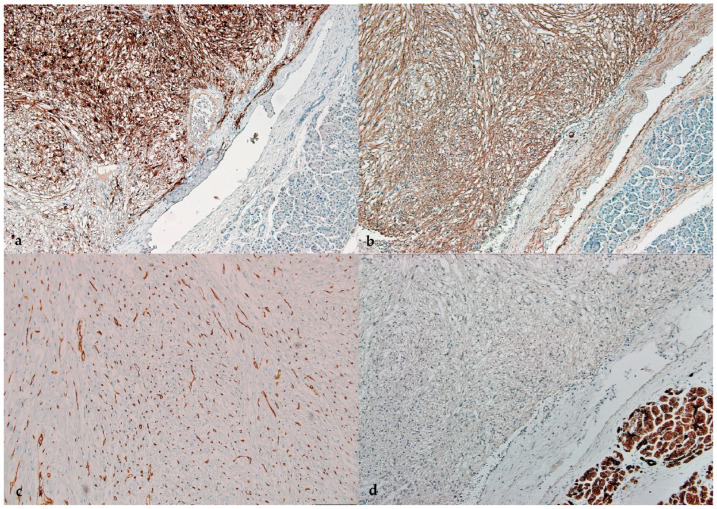
Immunohistochemical stains. HMB45s show positive staining of tumor cells. Normal pancreatic acini show negative staining (**a**); smooth muscle actine (SMA) shows positive staining tumor cells. Normal pancreatic acini show negative staining (**b**); CD34 demonstrating rich vascularity of the tumor, with perivascular epithelioid cells closely related to the capillaries (**c**); and pancytokeratin shows positive staining of normal pancreatic acini, and negative staining of tumor cells (**d**).

**Table 1 healthcare-11-00547-t001:** Reported Cases of PEComa of the Pancreas (1996–2020).

No.	Author, Year, (Reference)	Age/Sex	Symptoms	Location of Tumor in Pancreas	Size in cm	FNA	Operation	Margin	Histologic Type	Follow upMonths/Mets or Recurrence
1.	Zamboni, 1996 [4]	60/F	Abdominal pain	Body	2	Yes	DP	Free	Epithelioid Clear Cell Sugar Tumor	3/Negative
2.	Heywood, 2004 [13]	74/F	Abdominal pain	Uncinate process	4.7	No	PPPD	Free	Epithelioid Angiomyolipoma	69/Negative
3.	Ramuz, 2005 [14]	31/F	Abdominal pain	Body	1.5	Yes	SPDP	Free	Epithelioid Sugar Tumor	9/Negative
4.	Perigny, 2008 [15]	46/F	Diarrhea	Body	1.7	No	Excision	NAD	Epithelioid- Spindle PEComa	3/Negative
5.	Baez, 2009 [16]	60/F	Abdominal “bulge”	Body	3.5	Yes	DP	Free	Epithelioid- spindle PEComa (Sugar Tumor)	7/Negative
6.	Hirabyashi, 2009 [17]	47/F	Abdominal pain	Head	1.7	No	PPPD	NAD	Spindle PEComa	13/Negative
7.	Zemet, 2011 [18]	49/M	Fever, cough, malaise. History of Hodgkin lymphoma.	Head	4	Yes	PPPD	Free	Epithelioid- spindle PEComa	10/Negative
8.	Nagata, 2011 [19]	52/M	Abdominal pain	Head	4	No	PD	Positive	Malignant Epithelioid PEComa	39/Liver metastasis
9.	Xie, 2011 [20]	58/F	Upper abdominal pain, dyspepsia	Head	2.2	No	PD	Free	Epithelioid Angiomyolipoma (PEComa)	5/Negative
10.	Finzi, 2012 [21]	62/F	Incidental during follow up of liver hemangiomas	Posterior part of head	2.5	Yes	Total surgical resection	NAD	Epithelioid PEComa	5/Negative
11.	Al Haddad, 2013 [22]	38/F	Abdominal pain	Uncinate process	1.8	Yes	PD	NAD	Epithelioid PEComa	NAD
12.	Mourra, 2013 [23]	51/F	Jaundice, right hypochondriac pain, pruritus	Head	6	Yes	PD	Free	Malignant Epithelioid PEComa	6/Liver metastasis
13.	Okuwaki, 2013 [24]	43/F	Abdominal pain	Body	10	Yes	DP with splenectomy and partial gastrectomy	NAD	Spindle PEComa	7/Negative
14.	Kim, 2014 [25]	31/F	Incidental finding	Tail	3.3	No	DP	NAD	Angiomyolipoma	NAD
15.	Petrides, 2015 [26]	17/F	Melena, anemia	Head	4.2	Yes	PPPD	Free	Epithelioid and Spindle PEComa	18/Negative
16.	Kiriyama, 2016 [27]	57/M	Low back pain	Intrahepatic heterotopic pancreas	0.38	No	Left hemi-hepatectomy	Free	Spindle PEComa	56/Negative
17.	Mizuuchi, 2016 [28]	61/F	Abdominal pain	Head and body	6	No	PD	Free	Epithelioid PEComa	144/Negative
18.	Hartley, 2016 [29]	31/F	Incidental finding. History of tuberous sclerosis.	Tail. Three small sized PEComas, and other tumors	0.12, 0.2, and 0.6	Yes	DP for three PEComas and other tumors	Free	Epithelioid PEComas	NAD
19.	Jiang, 2016 [30]	50/F	Incidental finding on US	Head	2	Yes	PPPD	Free	Epithelioid PEComa	14/Negative
20.	Collins, 2017 [31]	54/F	Right upper quadrant abdominal pain	Body	2.6	Yes	MP	Free	Epithelioid PEComa	23/Negative
21.	Zhang, 2017 [32]	34/F	Abdominal pain	Head	11.5	No	PD and partial hepatectomy	NAD	Malignant Epithelioid PEComa	13/Negative Patient received chemotherapy
22.	Zizzo, 2018 [33]	68/M	Abdominal pain	Head	2.8	Yes	No surgery	NA	PEComa diagnosed via FNA	20/Alive with disease
23.	Sangiorgio, 2019 [34] Case 1	47/F	NAD	Isthmus	3	Yes	No surgery	NA	PEComa diagnosed via FNA	NAD
24.	Sangiorgio, 2019 [34] Case 2	70/F	NAD	Body	2	Yes	No surgery	NA	PEComa diagnosed via FNA	NAD
25.	Gondran, 2019 [35]	17/M	Weight loss, asthenia, and flu-like syndrome	Head	5.5	Yes	No surgery Treated with Sirolimus (mTOR inhibitor)	NA	Malignant Epithelioid PEComa	42/Marked reduction in tumor size
26.	Uno, 2019 [36]	49/F	Incidentally discovered on abdominal US	Tail	4.3	Yes	DP	Free	Spindle Cell PEComa	12/Negative
27.	Ulrich, 2020 [37]	49/F	Epigastric pain and intermittent diarrhea	Body	2.5	Yes	Left-sided pancreat-ectomy and splenectomy	Free	PEComa	NAD
28.	Rodríguez, 2020 [38]	50/F	Abdominal pain	Body	1.3	Yes	DP with splenectomy	NA	PEComa	18/Negative
29.	Almousa et al., 2023, (Current Case)	27/F	Epigastric pain	Tail	4.2	No	DP	Positive	Epithelioid PEComa	84/Negative

DP, distal pancreatectomy; MD, middle pancreatectomy; NA, not applicable; NAD, no available data; PD, pancreaticoduodenectomy; PEComas, perivascular epithelioid tumors; PPPD, pylorus-preserving pancreaticoduodenectomy; SPDP, splenic-preserving distal pancreatectomy; US, ultrasound.

**Table 2 healthcare-11-00547-t002:** Reported Cases of PEComa family in Patients with Ulcerative Colitis.

No.	Author, Year (Reference)	Age/Sex	Organ	Size cm	PEComa Family Type	Histologic Type	Treatment	Duration of UC	Follow Up
1.	Suzuki, 1996 [47]	38/Male	Liver	7.5	Angiomyolipoma	Classic	Hepatic lobectomy	12 years	Discharged in one month
2.	Prasad, 2000 [48]	22/Female	Cecum	3	Angiomyolipoma	Pleomorphic	Right hemi-colectomy	14 years	Postop bowel obstruction. Free for 5 months post treatment
3.	Leong, 2010 [49]	39/Male	Kidney	20	Angiomyolipoma	Classic	Embolization	NAD	30 months. Smaller size
4.	Sangiorgio, 2019 Case 4 [34]	41/Female	Liver, left lobe	1.6	PEComa	NAD	NAD	NAD	NAD
5.	Almousa, 2023 Current case	27/Female	Pancreas	4.2	PEComa	Spindle cell type	Distal pancreatectomy	11 years	84 months. No recurrence or metastasis

NAD, no available data; PEComa, perivascular epithelioid cell tumor.

## Data Availability

Not applicable.

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
