# Peer review of "Perivascular Epithelioid Cell Tumor (PEComa) of the Pancreas in a Patient with Ulcerative Colitis: A Case Report and Review of the Literature"

_healthcare, 2023, doi:10.3390/healthcare11040547_

Round 1

Reviewer 1 Report

Dear authors,

the manuscript "Perivascular Epithelioid Cell Tumor (PEComa) of the Pancreas in a Patient with Ulcerative Colitis: A Case Report and Review 3 of Literature" appears quite well written, the bibliography is appropriate and can represent a valid addition to the already existing bibliography.

Please find my comments:

- histology pictures should be marked with letters (e.g. a-b-c) and the figure caption should be better explained with reference to figure a,b or c.

- Immunohistochemistry images should be marked with letters (e.g. a-b-c) and the figure caption should be better explained with reference to figure a,b or c.

Reviewer 2 Report

General comments

- The case report title "Perivascular Epithelioid Cell Tumor (PEComa) of the Pancreas 2 in a Patient with Ulcerative Colitis: A Case Report and Review 3 of Literature" is very interesting in the field of cancer research. The study is not, however, novel. The articles that follow report on similar studies, and the uniqueness of the study should be addressed.

1. Perivascular epithelial cell tumor (PEComa) of the pancreas: A case report and review of literature - PubMed (nih.gov)

2. Perivascular epithelioid cell tumor (PEComa) of the pancreas: Immunoelectron microscopy and review of the literature - Hirabayashi - 2009 - Pathology International - Wiley Online Library

3. Perivascular epithelial cell tumor (PEComa) of the pancreas: a case report and review of previous literatures - PubMed (nih.gov)

4. Perivascular Epithelioid Cell Tumor (PEComa) of Pancreas Diagnosed Preoperatively by Endoscopic Ultrasound-Guided Fine-Needle Aspiration: A Case Report and Review of Literature - PubMed (nih.gov)

5. Pancreatic perivascular epithelial cell tumour (PECOma). Case report and literature review - PubMed (nih.gov)

6. Pancreatic perivascular epithelioid cell tumor: A case report with clinicopathological features and a literature review - PubMed (nih.gov)

Some specific comments

1. Images on Figures 2 and 3 are not labelled.

2. Some references are too old and is better to consider reference published for the last 5-10 years.

3. Better if the case report also includes the following validation tests: CD34, HMB45, SMA, additional protein quantifications, and mRNA.

Reviewer 3 Report

A well-written manuscript, which will be of interest to many in the field.

Major comments:

-          While performing the immunohistochemistry analysis, did the authors also check for any EMT markers like vimentin?

-          If the authors have access to the kidney function and liver function test results of the patient, can they make a table for that?

-          Did the patient have a history of smoking?
